# Multiple Targets of Toxicity in Environmental Exposure to Low-Dose Cadmium

**DOI:** 10.3390/toxics10080472

**Published:** 2022-08-13

**Authors:** Soisungwan Satarug, Glenda C. Gobe, David A. Vesey

**Affiliations:** 1Kidney Disease Research Collaborative, Translational Research Institute, Brisbane 4102, Australia; 2School of Biomedical Sciences, The University of Queensland, Brisbane 4072, Australia; 3NHMRC Centre of Research Excellence for CKD QLD, UQ Health Sciences, Royal Brisbane and Women’s Hospital, Brisbane 4029, Australia; 4Department of Nephrology, Princess Alexandra Hospital, Brisbane 4075, Australia

**Keywords:** cadmium, dietary exposure, fecundity, GFR loss, nephrotoxicity, selenium, toxicity threshold level, zinc

## Abstract

Dietary assessment reports and population surveillance programs show that chronic exposure to low levels of environmental cadmium (Cd) is inevitable for most people, and adversely impacts the health of children and adults. Based on a risk assessment model that considers an increase in the excretion of β_2_-microglobulin (β_2_M) above 300 μg/g creatinine to be the “critical” toxicity endpoint, the tolerable intake level of Cd was set at 0.83 µg/kg body weight/day, and a urinary Cd excretion rate of 5.24 µg/g creatinine was considered to be the toxicity threshold level. The aim of this review is to draw attention to the many other toxicity endpoints that are both clinically relevant and more appropriate to derive Cd exposure limits than a β_2_M endpoint. In the present review, we focus on a reduction in the glomerular filtration rate and diminished fecundity because chronic exposure to low-dose Cd, reflected by its excretion levels as low as 0.5 µg/g creatinine, have been associated with dose-dependent increases in risk of these pathological symptoms. Some protective effects of the nutritionally essential elements selenium and zinc are highlighted. Cd-induced mitochondrial dysfunction is discussed as a potential mechanism underlying gonadal toxicities and infertility.

## 1. Introduction

Cadmium (Cd) is a toxic metal of growing public health concern due to its widespread contamination of staple foods and air, and its exquisite toxicity to multiple organ systems [1,2,3]. Like all other metals, Cd persists indefinitely in the environment because of its nonbiodegradability. Despite its low levels in Earth’s crust and most soils, it can accumulate in vegetation because of its efficient soil-to-plant transference [4,5,6]. Cd is found in high abundance and is associated with the zinc (Zn) ores greenockite and sphalerite; thus, it is a byproduct of the mining, smelting, and refining of Zn ores, and has been used in many industrial processes [5,6]. Due to the realization of its high toxicity, the worldwide production and industrial uses of Cd have greatly reduced. However, the continued use of Cd-contaminated phosphate fertilizers still adds Cd to the food chain in most parts of the world [7,8,9].

As food crops form the major source of non-workplace Cd exposure in the non-smoking population [10,11], and because an outbreak of severe Cd poisoning, called “itai-itai” disease, revealed a health threat from the Cd contamination of rice [12,13], exposure guidelines and toxicity threshold levels of Cd in the human diet were established [14,15]. In addition, dietary assessment methods such as food-frequency questionnaires, duplicate diet studies and total diet studies (TDSs) have been used to monitor population exposure [10,11,16].

TDS is known also as the “market basket survey” because it involves the collection of samples of foodstuffs from supermarkets and retail stores for the quantitation of various food additives, pesticide residues, contaminants, and nutrients [10,11]. It is a reasonable method to identify sources as well as to gauge the levels of various contaminants and food additives in the human diet. Various food authority agencies such as the U.S. Food and Drug Administration (FDA), the European Food Safety Agency (EFSA), and the Food Standards of Australia and New Zealand (FSANZ) are tasked with conducting these monitoring programs. Overall, the TDS data indicate that Cd intake varies widely among populations, but the foods that are frequently consumed in large quantities, such as rice, potatoes, wheat, and leafy salad vegetables, are consistently the major sources of Cd [1,2].

Numerous studies suggest that exposure to Cd adversely impacts the functions of most organs of the body, in addition to its well-known toxicity in the kidneys and bones (itai-itai disease) and that a tolerable dietary exposure level is much lower than previously estimated. Thus, there is an urgent need to reevaluate the current health guidelines on its toxicity threshold and an acceptable (tolerable) consumption level. The present review provides an update on the toxicity of Cd in the kidneys and other organs, notably the gonads, and argues that the criteria by which the toxicity of Cd is judged should be reassessed.

## 2. Cadmium Tolerable Intake Level and Toxicity Threshold Level

The Joint FAO/WHO Expert Committee on Food Additives and Contaminants (JECFA) considered the kidney to be the critical target of Cd toxicity [14]. By definition, the provisional tolerable weekly intake (PTWI) for a chemical with no known biological function is an estimate of the amount that can be ingested weekly over a lifetime without an appreciable health risk. In 2010, the PTWI for Cd was amended to a tolerable monthly intake (TMI) of 25 μg per kg body weight per month, equivalent to 0.83 μg per kg body weight per day. Similarly, a Cd excretion rate of 5.24 μg/g creatinine was adopted as a nephrotoxicity threshold value [14].

The tolerable intake level derived by JECFA was based on a risk assessment model that considered an increase in the excretion rate of the low-molecular-weight protein β_2_-microglubulin (β_2_M) above 300 μg/g creatinine to be a “critical” endpoint. The European Food Safety Authority (EFSA) accepted the same endpoint. However, the EFSA designated a Cd excretion rate of 1 μg/g creatinine as the toxicity threshold with their inclusion of an uncertainty factor (safety margin), where an intake of 0.36 μg/kg body weight per day for 50 years was derived as an acceptable Cd ingestion level or reference dose (RfD) [15,16]. In theory, a threshold of toxicity is defined as the highest dose that does not produce an adverse effect in the most sensitive organ [17].

In a recent assessment, β_2_M excretion levels of 100–299, 300–999, and ≥1000 μg/g creatinine were associated with 4.7-, 6.2- and 10.5-fold increases in the risk of an estimated glomerular filtration rate (eGFR) ≤ 60 mL/min/1.73 m^2^, commensurate with CKD [18]. Thus, a cut-off value for an elevation of β_2_M excretion above 300 μg/g creatinine does not appear to be an early warning sign of the nephrotoxicity of Cd. The utility of β_2_M excretion as a toxicity criterion to derive a toxicity threshold level for Cd is questionable. A further discussion on β_2_M excretion as a marker of tubulopathy is provided in Section 4.4.

## 3. Organs Susceptible to Cadmium Toxicity

### 3.1. Fate of Cadmium in the Body

As Figure 1 depicts, ingested Cd is absorbed by the intestine and transported via the portal blood system to the liver, where its uptake induces the synthesis of metallothionein (MT) and the formation of CdMT complexes [19]. Later, hepatic CdMT is released into the systemic circulation. The fraction of absorbed Cd not taken up by hepatocytes in the first pass reaches systemic circulation and is taken up by tissues and organs throughout the body, including the kidneys, pancreas [20], ovaries [21] and testes [22].

The liver serves as an endogenous source of Cd^2+^ ions of dietary origin. From here, they are released and redistributed to kidneys as CdMT. In the circulation, less than 10% of Cd is present in plasma, and the remainder is in erythrocytes, where most Cd in whole blood is found. The whole-blood Cd level is indicative of recent exposure because the average lifespan of erythrocytes is 120 days [23].

In theory, Cd in non-MT forms can be taken up by all nucleated cells because they have the capacity to assimilate all the metals required for normal cellular metabolism and function. However, most cells do not take up CdMT because they lack the requisite mechanisms for protein internalization. Kidney proximal tubular epithelial cells provide an exception to this rule because of their capacity for receptor-mediated endocytosis, which facilitates the reabsorption of virtually all filtered proteins [24,25]. Filtered Cd in non-MT forms may be reabsorbed through many other nephron transporter systems, detailed in Section 4.2.

### 3.2. Target Organ Toxicity Identified from U.S. NHANES

As discussed in Section 2 above, an increase in β_2_M excretion above 300 μg/g creatinine was used as an endpoint in the health-risk assessment of Cd in the human diet, and urinary Cd excretion levels below 5.24 µg Cd/g creatinine were identified as the body burdens that were not associated with a change in β_2_M excretion [14]. Consequently, renal tubular dysfunction has become the most frequently reported adverse effect of environmental Cd exposure. However, many population-based studies in many countries and the U.S. general population study known as National Health and Nutrition Examination Survey (NHANES) have provided ample evidence that Cd exposure may impact the functions of many organ systems at Cd excretion levels below 5 µg/g creatinine.

NHANES is a cross-sectional study that has provided data on levels of exposure to more than 200 chemicals [26]. Urinary and blood Cd levels were quantified via a standardized methodology that enables the comparison of data across NHANES cycles [26]. The average Cd consumption estimated for the U.S. general population was 4.63 μg/d [27]. This figure was based on 24 h dietary recalls obtained for NHANES 2007–2012 participants aged 2 years and older (*n* = 12,523), plus the Cd levels of 260 food items in the 2006–2013 market basket surveys [27]. Cereals and bread, leafy vegetables, potatoes, legumes and nuts, stem/root vegetables, and fruits contributed to 34%, 20%, 11%, 7%, and 6% of total intake, respectively. Foods that contain relatively high Cd levels are spaghetti, bread, potatoes, and potato chips which contributed the most to total Cd intake, followed by lettuce, spinach, tomatoes, and beer. Lettuce was a main Cd source for White people and Black people. Tortillas and rice were the main Cd sources for Hispanic Americans and Asians plus other ethnicities [27].

The geometric mean, the 50th, 75th, 90th, and 95th percentile values for urinary Cd levels in the representative U.S. general population were 0.210, 0.208, 0.412, 0.678, and 0.949 µg/g creatinine, and the corresponding values for blood Cd were 0.304, 0.300, 0.500, 1.10, and 1.60 µg/L, respectively [28]. Based on the above figures for dietary exposure and urinary and blood Cd levels, environmental Cd exposure levels in the U.S. could be considered as low.

The urinary excretion of Cd and its blood levels associated with adverse effects on the kidneys [29,30,31,32], liver [33,34,35], and pancreas [36,37,38] are provided in Table 1.

The associations between Cd expsoure and reduced eGFR and albuminuria were consistently observed among participants in NHANES cycles undertaken between 1999 and 2016. A further analysis of these kidney outcomes is provided in Section 4.5. For liver outcomes, the hepatotoxicity of Cd in children appeared to be more pronounced in boys than girls [35]. In adults, urinary Cd levels ≥ 0.6 µg/g creatinine were associated with an increased risk of liver inflammation, NAFLD, and NASH in adults [33]. In Korean population studies, Cd hepatotoxicity was observed at blood Cd levels of 1–2 µg/L [39,40].

Urinary Cd levels of 1–2 µg/g creatinine were associated with increases in the risks of prediabetes and diabetes among U.S. adults [36,37]. Cd exposure was associated with an elevated risk of diabetes in a community-based study in Dallas, Texas [41]. A risk analysis using data from 4530 adults enrolled in NHANES 1999–2006, of which 10.3% had diabetes. Urinary Cd levels of 0.198 and 0.365 μg/g creatinine were found to be the benchmark dose (BMD) limit_5_ and BMDL_10_ for type 2 diabetes [38]. Thus, urinary Cd levels of 0.198 and 0.365 μg/g creatinine were the body burdens at which the prevalence of type 2 diabetes was likely smaller than 5% and 10%, respectively [38]. In a meta-analysis of pooled data from 42 studies, the risks of prediabetes and diabetes increased linearly with blood and urinary Cd; prediabetes risk reached a plateau at a urinary Cd of 2 µg/g creatinine, and the diabetes risk rose as blood Cd reached 1 µg/L [42]. These urinary Cd and blood Cd levels are also in range with those associated with reduced eGFR and albuminuria in studies conducted in the Torres Straits, Australia [43], Thailand [44], and Taiwan [45].

## 4. The Kidney as a Target of Cadmium Toxicity

### 4.1. Accumulation of Cadmium in Human Organs

Australian autopsy data show that renal cortical Cd content increases progressively until the age of 50 and declines thereafter [46,47]. The rates of Cd accumulation in the kidneys were 3–5 µg/g wet tissue for each 10-year increase in age, and the cortical content reached 25.9 µg/g wet tissue by 50 years of age. It is speculated that nephron loss and interstitial scarring due to aging and Cd toxicity caused the observed decline in cortical Cd content after age 50. Hepatic Cd content increased gradually with age without a sharp fall, as did kidney Cd content. Of interest, in a population-based study of Chinese subjects aged 2.8 to 86.8 years (*n* = 1235), urinary Cd excretion levels increased with age, peaking at 50 years in non-smoking women and 60 years in non-smoking men [48].

In a Swedish autopsy study, the average renal Cd content was 11 µg/g kidney wet weight for non-smokers aged 50 years [49]. Approximately 0.001–0.005% of Cd in the body was excreted in the urine each day, and the biological half-life of Cd in the kidney cortex was estimated to be 30 years for non-smokers [49,50]. The rate of kidney Cd accumulation among Swedish kidney-transplant donors who did not smoke was 3.9 μg/g kidney wet weight for every 10-year increase in age; this rate of kidney Cd accumulation rose to 4.5 μg/g kidney wet weight every 10 years in non-smoking women with low body-iron stores [51].

### 4.2. Reabsorption of Cadmium and Its Excretion

As Figure 2a,b depict, kidney tubular epithelial cells are well equipped to retrieve virtually all nutrients that pass the glomerular membrane into the filtrate. Under normal physiological conditions, blood perfuses the kidneys at a rate of 1 L per minute, all of which is directed through afferent arterioles into the glomeruli [52].

Owing to its small molecular weight, CdMT passes through the kidney glomerular membrane and is reabsorbed by tubular epithelial cells, and then, it is degraded in lysosomes [53,54,55,56,57]. The resultant free Cd^2+^ ions induce the synthesis of nascent MT to which Cd^2+^ ions are bound. Newly formed CdMT complexes are retained until released by injured or dying cells. Many other transporter systems for retrieving nutrients such as divalent metal transporter 1, organic cation transporters, transient receptor potential vanilloid 5, cysteine transporter, and the zinc (Zn) influx transporters ZIP8 and ZIP14 may also mediate the reabsorption of filtered non-MT-bound forms of Cd [54,58,59,60]. Eventually, the Cd absorbed by the gut and lungs is accumulated in the kidney tubular cells. The kidney Cd content is proportional to the amount assimilated from exogenous sources over a lifetime [51].

The kidney burden of Cd as µg/ g tissue weight increases with age [46,49]. An excretion of Cd can be viewed as the manifestation of ongoing Cd toxicity to kidney tubular cells because the excreted Cd emanates from injured or dying tubular cells [61], as does the enzyme N-acetyl-β-D-glucosaminidase (NAG) [62]. In Japanese residents of a Cd pollution area, the average half-life of the metal among those with a lower body burden (urinary Cd < 5 µg/L) was 23.4 years; in those with a higher body burden (urinary Cd > 5 µg/L), the average half-life was 12.4 years [63,64]. Thus, the lower the body burden, the longer the half-life of Cd. A half-life of 45 years was estimated from a Cd-toxicokinetic model that used data from Swedish kidney-transplant donors exposed to low environmental levels [65].

### 4.3. Kidney Tubular Cell Injury

The excretion of NAG [62,66] and kidney injury molecule 1 (KIM1) [67] have been used to quantify the injury to kidney tubular cells associated with Cd exposure [68,69]. Urinary NAG excretion is considered to be proportional to nephron numbers, as this enzyme mostly originates from tubular epithelial cells and is released upon cell injury [62]. In a United Kingdom (U.K.) study, a dose–response relationship was observed between urinary Cd and NAG levels [68]. Furthermore, a urinary Cd of 0.5 μg/g creatinine was associated with 2.6- and 3.6-fold increases in the prevalence of a urinary NAG > 2 U/g creatinine, as compared with a urinary Cd of 0.3 and <0.5 μg/g creatinine, respectively [68]. Like NAG, KIM1 emanates from tubular cells, and its excretion is correlated with that of Cd [70,71]. KIM1, the most sensitive marker of the toxicity of Cd in the kidneys, is found in the urine only after tubular injury has occurred [67,71]. In addition, an association between the excretion of Cd and KIM1 excretion was noted in Taiwanese subjects with CKD after adjusting for covariates [45]. Interestingly, no correlation was found between the excretion of Cd and protein, thereby suggesting that urinary KIM-1 levels could serve as an early warning sign of kidney injury due to low-dose Cd exposure. In a Guatemalan study, Cd excretion was associated with increased excretion of neutrophil gelatinase-associated lipocalin, another sensitive marker of kidney tubular cell injury [72].

### 4.4. Impaired Reabsorption of Filtered Proteins

β_2_M, with a molecular weight of 11,800 Da, is synthesized and shed by all nucleated cells [73]. By virtue of its small mass, β_2_M is filtered freely by glomeruli, and is reabsorbed almost completely by proximal tubular cells [74]. Cd has been shown to cause a reduction in a tubular maximum (Tm) for β_2_M reabsorption [75], and for many decades, investigators have measured β_2_M excretion to document Cd-induced reabsorptive dysfunction [76,77,78]. Several studies of Cd-exposed subjects have demonstrated inverse relationships between eGFR and β_2_M excretion [18,74,79,80,81,82], but none have quantified the individual contributions of low eGFR and reduced biomarkers for reabsorption to increased β_2_M. It is noteworthy that there are attributes of β_2_M excretion that compromise its utility as a marker of tubulopathy. β_2_M production rises in response to many inflammatory and neoplastic conditions [79]. If reabsorption rates of β_2_M per nephron remain constant as its production rates change, excretion will vary directly with its production. If the production and reabsorption per nephron remain constant as nephrons are lost, the excretion of β_2_M will rise.

### 4.5. Reduced Glomerular Filtration Rate

Reductions in GFR due to Cd nephropathy have been attributed to glomerular injury. Although this inference may be at least partially correct, it is not necessary. Sufficient tubular injury disables glomerular filtration and ultimately leads to nephron atrophy, glomerulosclerosis, and interstitial inflammation and fibrosis [83]. The GFR is considered the best indicator of nephron function because it reflects the number of surviving nephrons × the average GFR per nephron [84]. However, measurement of the GFR is not feasible in population-based studies. In practice, the GFR is estimated from equations and is reported as an estimated GFR (eGFR).

A quantitative analysis of the excretion of Cd and NAG suggests that Cd inflicts tubular cell injury at low intracellular concentrations, and that toxicity intensifies as its concentration rises [61]. Inflammation and fibrosis follow, nephrons are lost, and the GFR falls Figure 1, [61]. In a histopathological examination of kidney biopsies from healthy kidney transplant donors [85], the degree of tubular atrophy was positively associated with the level of Cd accumulation. The tubular atrophy was observed at relatively low Cd levels (median 13 µg/g wet tissue weight) [85].

In a prospective cohort study of Bangladeshi preschool children, an inverse relationship between urinary Cd excretion and kidney volume was seen in children at 5 years of age. This was in addition to a decrease in eGFR [86]. Urinary Cd levels were inversely associated with eGFR, especially in girls, and such an inverse association between eGFR and urinary Cd was particularly strong in those with urinary selenium (Se) below 12.6 μg/L, thereby suggesting a protective effect of Se [86].

In another prospective cohort study of Mexican children, the reported mean for Cd intake at the baseline was 4.4 µg/d, which rose to 8.1 µg/d after nine years, when such Cd intake levels showed a marginally inverse association with eGFR [87].

In a Korean cross-sectional study, blood Cd levels in the highest tertile were associated with 1.85 mL/min/1.73 m^2^ lower eGFR values, compared with the lowest tertile [88]. Higher Cd excretion was associated with lower eGFR and was a sign of tubular injury in a study of Guatemalan sugarcane workers [72] and Myanmarese subjects [89]. Both the tubular and glomerular effects were observed in a study of Swedish women, despite the exposure levels being low; urinary Cd 0.67 µg/g creatinine was positively associated with NAG, while urinary Cd 0.87 µg/g creatinine was inversely associated with eGFR [90].

Environmental Cd exposure in the U.S was associated with an approximate 2-fold increase in the risk of reduced eGFR (Table 1). Furthermore, a protective effect of Zn was observed in an analysis of data from 1545 participants aged ≥ 20 years in NHANES 2011–2012 [31]. Blood Cd levels > 0.53 μg/L were associated with a 2.04-fold increase in the risk of low eGFR, compared with blood Cd levels < 0.18 μg/L. The risk of CKD rose to 3.38 in those who had serum Zn levels < 74 μg/dL.

### 4.6. Cadmium Exposure and Chronic Kidney Disease

CKD is defined as an eGFR ≤ 60 mL/min/1.73 m^2^ or albuminuria (a urinary albumin-to-creatinine ratio above 30 mg/g creatinine) that persists for at least 3 months [91]. Cd exposure has been associated with reduced eGFR and albuminuria among U.S. adults enrolled in the 1999–2016 NHANES cycles (Table 1). In a Spanish population study, a urinary Cd level as low as 0.27 µg/g creatinine was associated with an increase in the risk of albuminuria by 58% [92].

In a cross-sectional analysis, urinary Cd levels > 1.72 µg/g creatinine were associated with elevated albumin excretion among environmentally exposed Shanghai residents [93]. In a Chinese population study, dietary Cd intake estimates at 23.2, 29.6, and 36.9 μg/d were associated with 1.73-, 2.93- and 4.05-fold increments in the prevalence of CKD, compared with the 16.7 μg/d intake level [94]. A diet high in rice, pork, and vegetables was associated with a 4.56-fold increase in the prevalence of CKD [94].

Blood Cd concentrations of 1.74 μg/L and 2.08 μg/L were associated with low eGFR in cross-sectional studies of a Korean population [95,96]. These blood Cd levels were approximately 4 times higher than blood Cd levels, which was found to be associated with low eGFR in a representative U.S. population [32,33,34]. An inverse association was seen between blood Cd and eGFR in another study of Koreans aged ≥ 19 years [97].

Although studies from various countries report disparate levels of urinary and blood Cd, they are broadly consistent in that they find that urinary Cd levels associated with low GFR values and albuminuria did not exceed a prescribed Cd toxicity threshold level of 5.24 µg/g creatinine [14]. This Cd toxicity threshold level was derived from a risk assessment model that considers β_2_M excretion above ≥ 300 µg/ g creatinine to be the critical endpoint. However, the data presented above indicate that an increase in β_2_M excretion is not sensitive enough to observe pathological changes below a prescribed toxicity threshold level. Reassessment of the current guidelines on dietary exposure limits and of the threshold for the nephrotoxicity of Cd are warranted.

Cd-induced GFR reduction is one of the endpoints suitable for health-risk calculation, although data from longitudinal studies are presently lacking. This endpoint is clinically relevant, and importantly, this effect has been observed in children and adults. In clinical trials, the successful treatment of kidney disease is judged by the attenuation of a decline in eGFR [91,98].

Health-risk assessment should be based on the organ most sensitive to the toxicity of Cd, and on subpopulations with increased susceptibility to Cd toxicity such as children. CKD is a progressive syndrome with high morbidity and mortality and affects 8% to 16% of the world’s population [91,98]. Of concern, the incidence of CKD continues to rise globally. Given the escalating treatment costs associated with dialysis and/or kidney transplants needed for survival, the minimization of environmental pollution and population exposure to Cd are important preventive measures, as are other risk-reduction measures.

## 5. The Gonad as a Target of Cadmium Toxicity

Infertility affects approximately 15% of childbearing-age couples worldwide, and an evolving body of evidence suggests low environmental exposure to Cd as one of the contributing factors [99,100]. Sapra et al. (2016) observed an effect of low-dose Cd exposure on fecundity from a population-based prospective cohort study of 501 couples in Michigan and Texas [101]. In this cohort study, higher blood Cd levels were associated with diminished fecundity, measured as a long time-to-pregnancy (TTP), the number of menstrual cycles or months of unprotected intercourse required to achieve pregnancy. Fecundity was reduced by 59% and 47% in men and women, respectively. Cigarette smoking was found to also be associated with longer TPP, but the strength of this association was attenuated after adjusting for blood Cd levels. Thus, Cd may partially mediate the effect of smoking on long TTP in women and men who smoked.

### 5.1. Measures of Ovarian Reserves and Testicular Health

The female gametes (oocytes) and male gametes (spermatozoa) are produced by the ovaries and testes. In addition, various hormones, including estrogen, testosterone, progesterone, anti-Müllerian hormone (AMH), and insulin-like peptide 3 (INSL3), are produced here [102,103,104]. In women, the serum concentration of AMH reflects the ovarian pool status [105,106,107]. In men, testosterone suppresses the testicular synthesis of AMH by Sertoli cells, while the synthesis and secretion of INSL3 by Leydig cells are independent of the hypothalamic–pituitary–gonad (HPG) axis [103,108,109,110]. Thus, AMH and INSL3 are markers of Sertoli and Leydig cells’ maturation in men. Ovarian and testicular gametogenesis are activated by the follicle-stimulating hormone (FSH), and luteinizing hormone (LH) which are derived from the pituitary gland in response to the hypothalamic gonadotropin-releasing hormone (GnRH) [103,111,112].

The ovary contains a finite number of primordial follicles from which oocytes can be produced upon stimulation by FSH and LH [106,107]. The development of the primordial follicle into primary, secondary, antral, and Graafian follicles occurs in a cyclical manner. In each cycle (21–42 days), only one follicle undergoes complete maturation. The granulosa cells of developing follicles synthesize estrogen, while the granulosa cells of secondary, preantral, and early antral follicles produce AMH. The mature Graafian contains an oocyte, which is released during ovulation, and the remnant of the Graafian follicle becomes the corpus luteum, where progesterone is produced.

In contrast to the finite number of female oocytes that can be produced from primordial follicles, male primordial germ cells in the seminiferous tubules of the testes develop into millions of sperm cells throughout life [111,112,113]. The development of male germ cells into spermatids, and then spermatozoa, are nurtured by Sertoli cells and Leydig cells. Sertoli cells are nondividing cells, and they are active for the reproductive lifetime by changing their morphology and gene expression in a cyclical manner. Each Sertoli cell supports approximately 30–50 germ cells in different stages of development [108]. An estimated average duration for sperm maturation in humans is 74 days (with a range of 69–80) [113].

### 5.2. Cadmium and Zinc Levels in Gonads

An analysis of Cd contents of the ovaries of 94 Hungarian women, aged 16–76 yrs, who underwent bilateral adnexectomy reported a positive correlation of age with ovarian Cd content. In the 30–65 yr age-group, the Cd contents ranged between 0.02 and 0.2 µg/g wet weight [21]. An analysis of Cd in various organs from 41 Norwegian men (median age 40 yrs, range 18–80) reported mean values for the Cd content of the kidneys, liver, epididymis, seminal vesicles, prostate gland and testes as 93.7, 4.06, 0.97, 0.71, 0.63, and 0.56 µg/g dry weight, respectively [22]. The epididymis accumulated a greater Cd quantity than did the prostate gland and testes; however, the Cd content of the epididymis, prostate gland and testes all correlated more closely with age than did kidneys. Of interest, the kidney Cd contents correlated positively with Cd in the seminal vesicles and epididymis [22]. The levels of Cd and Zn found in seminal plasma samples reported in recent studies are provided in Section 5.3.

### 5.3. Cadmium Exposure Levels and Reproductive Health Outcomes

Table 2 provides a summary of the findings from epidemiological studies linking low levels of Cd in blood, urine, and semen samples to various indicators of reproductive health outcomes in women [114,115,116,117,118] and men [119,120,121,122]. The Cd toxicities to the testes are inferred from studies in which semen quality was examined according to the criteria prescribed by the WHO [123].

Lee et al. (2020) observed a dose–response relationship between blood Cd levels and infertility among women enrolled in NHANES 2013–2014 and 2015–2016 [114]. Blood Cd levels of 0.20–0.33 and 0.34–5.14 µg/L were associated with a 1.15- and 2.47-fold increased risk of infertility, compared with blood Cd levels ≤ 0.19 µg/L.

Among women enrolled in NHANES 1988–1994, Upson et al. (2021) observed a dose–response relationship between urinary Cd concentrations and evidence for the depletion of ovarian reserves (serum concentrations of FSH ≥10 IU/L [115]. Urinary concentrations of 0.16–0.38, 0.39–0.77, and >0.77 µg/L were, respectively, associated with a 1.4-, 1.6-, and 1.8-fold increased risk of ovarian reserve depletion, compared with urinary Cd levels below 0.16 µg/L.

Pan et al. (2021) observed an association between Cd excretion rate and the risk of primary ovarian insufficiency in Chinese women, defined as serum FSH levels > 25 IU/L [116]. Cd excretion rates ≥ 0.68 µg/g creatinine were associated with a 2.5-fold increased risk of ovarian insufficiency, compared with Cd excretion rates below <0.37 µg/g creatinine. In addition, inverse associations were seen between Cd excretion rate and serum concentrations of estradiol and AMH, an indicator of premature ovarian failure [116]. Likewise, Lee et al. (2018) reported an inverse association of blood Cd and serum AMH concentration, seen in Korean women, which was particularly strong in the 30–35-year age group [117].

An inverse correlation between semen Cd levels and sperm motility was reported by Mitra et al. (2020) in their study of Indian men, tea-garden workers included, who attended the fertility clinics of Southern Assam, India [86]. A positive correlation was noted between semen Cd levels and the percentages of morphologically abnormal sperm [118].

A tendency for associations between semen Cd levels and total sperm count and sperm motility was noted in a study of Italian men living in two areas of eastern Sicily [119]. The median Cd level in semen samples from men with abnormal sperm parameters were 2.17 times higher (1.43 vs. 0.66 μg/L), while the median semen selenium (Se) level was 10.6 times lower, compared to the median Cd and Se values in those with normal semen quality parameters. An interaction analysis suggested that Se may mitigate Cd effects on sperm concentration, total sperm count, and progressive sperm motility. In an analysis of Se and Zn in human reproductive organs, Se levels were the highest in the testes, and the prostate gland had the highest level of Zn [124]. The results from a study on mice suggested that the high level of Se in the testes may be required to stimulate testosterone synthesis in Leydig cells [125].

Wang et al. (2017) related semen Cd and Zn levels to the quality of semen samples from Chinese men who underwent investigation for subfertility at the Wuhan Reproductive Medicine Center [120]. They observed inverse associations between seminal plasma Cd levels and sperm motility (progressive and total motility). Theses associations persisted when other elements were included in their statistical model analysis. In addition, a positive association was seen between seminal plasma Zn levels and sperm concentrations [120]. Compared with seminal plasma Zn quartile 1, sperm concentrations rose by 13%, 23%, and 25% in the seminal plasma Zn quartiles 2, 3, and 4, respectively.

The protective effect of Zn may be due a reduction in testicular Cd accumulation. The ability of Zn to reduce the cellular uptake of Cd has been shown using human embryonic kidney cells (HEK293) overexpressing ZIP1 or ZIP8 [126,127]. ZIP8 is a Zn influx transporter involved in Cd-induced injury to the testes [128,129]

Jeng et al. (2015) observed an association between the Cd excretion rate and a reduction in sperm viability in Taiwanese men who underwent annual health examination at a main municipal hospital in the southern region of Taiwan [121]. Cd excretion rates ≥ 0.8 μg/g creatinine were associated with sperm viability ≤ 58%. Sperm viability, motility, and/or concentrations were inversely associated with urinary Cd, BMI, and age.

In their study of men in Hong Kong, Shi et al. (2021) found that high blood Cd levels (>1.44 μg/L) were associated with a decrease in the sperm acrosome reaction, which is critical in fertilization [122]. Of interest, blood iron levels showed a similar relationship with sperm functionality, measured as the acrosome reaction, but blood lead (Pb), Zn, molybdenum (Mo), and calcium (Ca) did not [122]. No abnormalities were seen in any of the other semen quality parameters.

The collective evidence detailed above suggests that low environmental exposure to Cd does affect human fecundity, a previously ill-recognized health problem. Urinary Cd levels ≥ 0.68 µg/g creatinine were associated with a 2.5-fold increase in the risk of ovarian insufficiency. Urinary Cd levels ≥ 0.8 μg/g creatinine were associated with a reduction in sperm viability. The relatively low body burden of Cd associated with Cd excretion rates of 0.68 μg/g creatinine and 0.8 μg/g creatinine are, respectively, 13% and 15% of the conventional threshold limit of 5.24 μg/g creatinine. Hence, the current toxicity threshold level is not protective of reproductive health.

### 5.4. Experimental Evidence for Gonadotoxicity of Cadmium

The recent data from experimental studies are summarized below to identify the potential mechanism(s) underpinning the effects of Cd exposure on human fecundity. However, it is noteworthy that relatively high Cd-dose regimes were used in most experimental studies to obtain measurable effects.

#### 5.4.1. Studies Using Cell Lines

Cd is a potent mitochondrial toxicant capable of uncoupling adenosine triphosphate (ATP) synthesis, inhibiting the electron transport chain, and promoting reactive oxygen species (ROS) formation with resultant oxidative stress conditions [130,131,132,133]. The organs with high metabolic activities and high energy demands, such as ovaries and testes, are particularly susceptible to the toxicants that affect mitochondrial function. Excessive production of ROS and the resultant oxidative stress may contribute to poor semen quality and infertility in men [134].

A reduction in testosterone synthesis in the Leydig cells of the testes by Cd is linked to its mitochondrial effects [135,136]. In an in vitro study using the rat Leydig tumor cell line R2C, Cd caused a decrease in mitochondrial membrane potential, reduced cAMP production, and diminished progesterone secretion [135]. A reduction in progesterone synthesis in R2C cells was linked to the suppression of the mitochondrial expression of the dihydrolipoamide dehydrogenase enzyme by Cd [136]. The knockdown of dihydrolipoamide dehydrogenase gene expression reduced progesterone synthesis by 40%, as did Cd treatment [136].

Likewise, the cytotoxicity of Cd in the human granulosa-like tumor (KGN) cell line, a cell culture model of human ovarian granulosa cells, has been linked to its toxicity toward mitochondria [137]. This KGN human granulosa-like cell line expressed the FSH receptor and retained progesterone synthesis and Fas-mediated apoptosis, like those of normal human ovarian granulosa cells [138]. Cd reduced mitochondrial membrane potential, leading to less ATP synthesis and enhanced ROS formation in KGN cells [137]. Elevated mitochondrial ROS and oxidative stress may accelerate ovarian aging, leading to the premature depletion of the ovarian pool, as discussed in Section 5.3.

#### 5.4.2. Ovarian Toxicity Studies in Whole Animals

The ovaries of rats given i.p. Cd at 0, 0.25, 0.5, or 1 mg/kg for 5 days/week for 6 weeks released less estradiol and progesterone [139]. The ovaries of Kunming mice given i.p. Cd at 6 mg/kg/body weight for 8 days had few primordial follicles, more atresia follicles, granulosa cells with abnormal morphology, and evidence of damage to their oocyte nuclei [140]. Notable ovarian effects in rats given i.p. Cd acetate at 0.05 mg/kg for 4 weeks included reduced serum estradiol levels, a prolonged estrous cycle, and no mature follicles [141]. A 46.7% decrease in serum AMH concentrations and few preantral and antral follicles with increased atretic follicle numbers were noted in Wistar rats given Cd in their drinking water (100 µg/L) for 30 days [142]. This experimental treatment produced ovaries with Cd contents of 1–4.5 µg/g wet weight and blood Cd concentrations of 1–4 µg/L, which were found to be inversely correlated with serum AMH levels [142]. In Wistar rats given Cd via gavage at 0.09 mg/kg body weight for 90 days, Cd raised the uterus estradiol contents, increased the endometrial epithelium thickness, and prolonged the estrus phase. These effects were like those of 17β-estradiol, and they persisted up to six months after 90 days of Cd treatment [143].

As detailed above, failure of follicular development, follicular atresia, and a prolonged estrus cycle were the effects of Cd dosing in female rats and mice. It is more likely that the effects were due to a reduction in FSH and LH produced by the pituitary gland, diminished follicular synthesis, and the release of progesterone, estrogen, and AMH. In addition, Cd may accelerate ovarian aging, leading to few primordial and primary follicles; this is indicative of ovarian reserve depletion. At low levels, Cd may mimic an effect of estradiol, leading to an increment in endometrial epithelial thickness.

#### 5.4.3. Testicular Toxicity Studies in Whole Animals

A marked reduction in serum testosterone was observed in C57BL/6J mice treated i.p. with Cd at 1.0 mg per kg of body weight for 1 week [144]. The reduction in serum testosterone was attributed to the enhanced degradation of heme through the induction of heme oxygenase-1 (HO-1), a well-known effect of Cd as a potent HO-1 inducer. Consequently, iron was released from heme, triggering ROS formation, lipid peroxidation and the loss of Leydig cells [144]. In another study, a decrease in serum testosterone concentration was noted together with seminiferous epithelial degeneration and germinal epithelial detachment in the testes of mice given a single oral dose of Cd at 30 mg/kg or a fractional oral dose of Cd at 4.28 mg/kg per day for 7 consecutive days [145]. A fractional oral Cd-dose regime produced higher testicular Cd accumulation and more extensive damage to the seminiferous epithelium than did a single bolus oral Cd [145]. In the testes of mice, the degeneration of the seminiferous epithelium, the death of germ cells, and damage to Leydig cells were seen 42 days (one seminiferous cycle in mice) after a single oral dose of Cd at 24 mg Cd/kg or a single i.p. dose of Cd at 1.2 mg Cd/kg [145]. A decrease in testicular Zn content was seen only in mice receiving oral Cd.

The duration of spermatogenesis (a seminiferous cycle) was estimated to be 56 days in rats and 42 days in mice [146,147]. A reduction in the volume of Leydig cells was observed 56 days after a single i.p. Cd dose at 0.34 mg/kg body weight [147]. In another experiment using rats, a single i.p. Cd dose at 0.5 or 1.0 mg/kg affected the differentiation of Leydig cells, leading to an irreversible loss in their regeneration, and a decrease in serum testosterone and LH measured at 41, 55, and 76 days after exposure [148]. Cd-induced germ cell death has been linked to autophagy in Sertoli cells in rats given a single i.p. dose of Cd at 2 mg/kg, which produced serum Cd concentrations of 4–10 µg/L, and a testicular Cd content of 0.2–0.3 µg/g wet weight [149].

## 6. Conclusions

The tolerable intake of Cd and its corresponding toxicity threshold at a urinary excretion of 5.24 µg/g creatinine were derived from a risk assessment model that considered an increase in urinary β_2_M excretion above 300 µg/g creatinine to be a critical effect. However, multiple organs, including the kidneys, liver, pancreas and gonads, are subjected to the cytotoxicity of Cd at body burdens much lower than previously estimated. Thus, the Cd exposure limit derived from a β_2_M endpoint is deemed unreliable, and is not sufficiently restrictive to be without a health risk. KIM1 has emerged as an early warning sign of Cd nephrotoxicity, and its use in conjunction with β_2_M, albumin, and eGFR in a longitudinal study design warrants further research. Clinically relevant pathological symptoms such as a reduction in eGFR and a decline in fecundity could serve as endpoints to derive a consumption level of Cd that carries a negligible health risk.

Low environmental Cd exposure adversely affected the function of the kidneys and liver in both children and adults. Information on the protective roles of Zn and Se against a reduction in eGFR, and on the quality of semen associated with Cd exposure, have emerged. As Cd raises the risk of such highly prevalent ill-health conditions as CKD, liver disease, and infertility, even small relative increases in the risk of these conditions suggest that large absolute numbers of cases may be preventable by reducing exposure to Cd, together with the adequate consumption of Zn and Se.

## Figures and Tables

**Figure 1 toxics-10-00472-f001:**
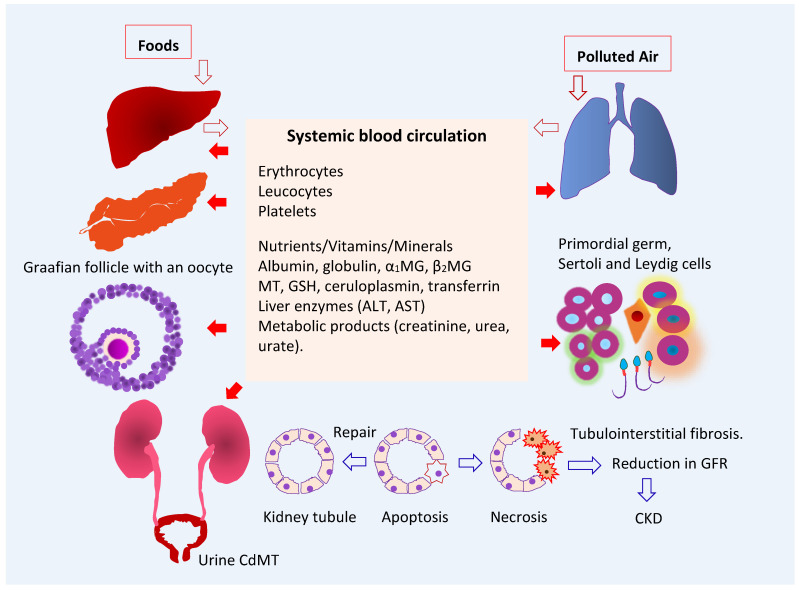
Multiple toxicity targets of cadmium. Ingested Cd is absorbed and transported to liver, where synthesis of MT is induced, and CdMT is formed. The fraction of absorbed Cd not taken up by hepatocytes in the first pass reaches systemic circulation and is taken up and accumulated by cells throughout the body. After glomerular filtration, CdMT is reabsorbed by kidney tubular cells. Other forms of filtered Cd can be reabsorbed by the kidney nephron transporters for iron, zinc, manganese, and calcium. Abbreviations: Cd—cadmium; MT—metallothionein; CdMT—cadmium-metallothionein complex; α_1_MG—α_1_-microgloulin; β_2_MG—β_2_-microglobulin; GSH—glutathione; ALT—alanine aminotransferase; AST—aspartate aminotransferase; GFR—glomerular filtration rate; CKD—chronic kidney disease.

**Figure 2 toxics-10-00472-f002:**
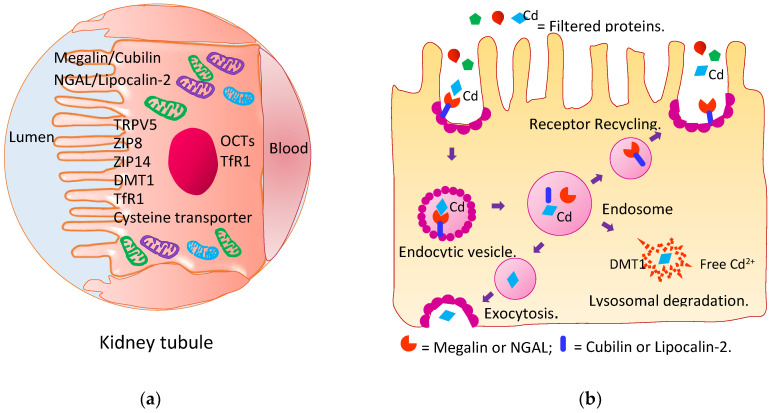
Metal transporters and receptors involved in cadmium reabsorption. (**a**) Metal transporters and receptors in the kidney tubule; (**b**) receptor-mediated endocytosis of filtered proteins. Reabsorption of Cd and CdMT by kidney tubular epithelial cells are mediated by multiple transporter systems such as the megalin/cubilin and NGAL/lipocalin 2 systems and transporters for zinc (Zn), iron (Fe), calcium (Ca) and manganese (Mn).

**Table 1 toxics-10-00472-t001:** Kidney, liver and pancreas as targets of toxicity to chronic exposure to low-dose cadmium.

Targets	NHANES Dataset	Adverse Effects and Risk Estimate	References
Kidneys	1999–2006	Blood Cd levels >1 µg/L were associated with low GFR ^a^ (OR 1.48) and albuminuria ^b^ (OR 1.41). The OR for albuminuria was increased to 1.63 in those with urinary Cd ≥ 1 µg/g creatinine plus blood Cd > 1 µg/L.	Ferraro et al. 2010 [29]
Kidneys	1999–2006	Blood Cd levels ≥ 0.6 μg/L were associated with low GFR (OR 1.32), albuminuria (OR 1.92) and low GFR plus albuminuria (OR 2.91).	Navas-Acien et al. 2009 [30]
Kidneys	2011–2012	Blood Cd levels ≥ 0.53 μg/L were associated albuminuria (OR 2.04) and low GFR (OR 2.21).	Lin et al. 2014 [31]
Kidneys	2007–2012	Blood Cd ≥ 0.61 μg/L were associated with low GFR (OR 1.80) and albuminuria (OR 1.60).	Madrigal et al. 2019 [32]
Liver	1988–1994	Urinary Cd levels ≥ 0.83 μg/g creatinine were associated with liver inflammation in women (OR 1.26). Urinary Cd ≥ 0.65 μg/g creatinine were associated with liver inflammation (OR 2.21), NAFLD (OR 1.30), and NASH (OR 1.95) in men.	Hyder et al. 2013 [33]
Liver	1999–2015	A 10-fold increment of urinary Cd was associated with elevated plasma levels of total bilirubin (OR 1.20), ALT (OR 1.36), and AST (OR 1.31).	Hong et al. 2021 [34]
Liver	1999–2016	A urinary Cd quartile 4 was associated with elevated plasma ALT (OR 1.40) and AST (OR 1.64). The effect was larger in boys than in girls.	Xu et al. 2022 [35]
Pancreas	1988–1994	Urinary Cd levels 1–2 μg/g creatinine were associated prediabetes (OR 1.48) and diabetes (OR 1.24).	Schwartz et al. 2003 [36]
Pancreas	2005–2010	Urinary Cd levels ≥ 1.4 µg/g creatinine were associated with pre-diabetes in non-smokers. In a fully adjusted model including smokers and non-smokers, urinary Cd levels between 0.7 and 0.9 µg/g creatinine were associated with pre-diabetic risk.	Wallia et al. 2014 [37]
Pancreas	1999–2006	Urinary Cd levels of 0.198 and 0.365 μg/g creatinine were identified as exposure levels at which the prevalence of type 2 diabetes was smaller than 5% and 10%, respectively.	Shi et al. 2021 [38]

NHANES—National Health and Nutrition Examination Survey; ^a^ low GFR is defined as estimated glomerular filtration rate (eGFR) ≤ 60 mL/min/1.73 m^2^; ^b^ albuminuria is defined as urinary albumin-to-creatinine ratio ≥ 30 mg/g. OR—odds ratio; NAFLD—non-alcoholic fatty liver disease; NASH—non-alcoholic steatohepatitis.

**Table 2 toxics-10-00472-t002:** Blood and urinary cadmium levels associated with increases in risk of adverse reproductive outcomes.

Endpoints	Dataset	Risk Estimate	References
Infertility	NHANES 2013–2014, 2015–2016	A 2-fold increment in blood Cd level was associated with infertility (OR 1.84). Blood Cd range: 0.07–5.14 µg/L.	Lee et al. 2020 [114]
Ovarian reserve depletion	NHANES 1988–1994	Urinary Cd levels > 0.77 µg/L were associated with serum follicle-stimulating hormone (FSH) levels ≥10 IU/L, indicative of ovarian reserve depletion (OR 1.8).	Upson et al. 2021 [115]
Ovarian insufficiency	Chinese (Zhejiang) women, n = 378	Urinary Cd levels and > 0.68 µg/g creatinine were associated with serum FSH levels ≥25 IU/L, indicative of primary ovarian insufficiency (OR 2.50).	Pan et al. 2021 [116]
Ovarian failure	Korean (Soul) women, n = 283	Blood Cd levels were inversely associated with serum anti-Mullerian hormone (AMH) levels, especially in 30–35-year age-group (β = −0.43) (*p* = 0.01).	Lee et al. 2018 [117]
Sperm motility	Indian (Assam) men, N = 400.	Semen Cd levels were inversely correlated with sperm motility (*r* = −0.987, *p* < 0.001). The percentage of morphologically abnormal sperm increased with semen Cd levels (*r* = 0.378, *p* < 0.001).	Mitra et al. 2020 [118]
Semen quality	Italian (Sicily) men, n = 179.	Semen Cd levels in men with abnormal sperm quality were 1.43 μg/L, 2.17 times higher than in those whose semen quality was normal.	Calogero et al. 2021 [119]
Sperm quality	Chinese (Wuhan), n = 746.	Semen Cd levels were inversely associated with progressive motility and total motility. Sperm concentration increased with semen Zn levels.	Wang et al. 2017 [120]
Sperm viability	Taiwanese men, n = 196.	Urinary Cd levels ≥ 0.8 μg/g creatinine were associated with sperm viability lower than 58%.	Jeng et al. 2015 [121]
Acrosome reaction	Hong Kong men, n = 288.	Blood Cd levels >1.44 μg/L were associated with a decrease in sperm acrosome reaction. The median blood Cd was 0.36 μg/L.	Shi et al. 2021 [122]

NHANES—National Health and Nutrition Examination Survey; OR—odds ratio. Fecundity was measured as time-to-pregnancy. Semen quality was examined according to the criteria prescribed by the WHO [123]. The urinary Cd excretion rates found to be associated with increases in risk of fecundity decline all were lower than the urinary Cd threshold level of 5.24 µg/g creatinine [14].

## Data Availability

Not applicable.

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
