# Peer review of "Multiple Targets of Toxicity in Environmental Exposure to Low-Dose Cadmium"

_toxics, 2022, doi:10.3390/toxics10080472_

Round 1
Reviewer 1 Report
The manuscript is generally well-written and thorough with respect to reproductive organ review. However, there are major concerns regarding completeness of the kidney review and conclusions /scientific premises that are not substantiated (see specific comments regarding lines 85/439 and 188). The section 4.2 on reduced eGFR should also include B2M (and other kidney injury markers) as outcomes. The citation of a single study in reference 18 (Satarug et al 2019) does not sufficiently or justifiably support the authors’ conclusions that B2M is not an appropriate biomarker in the sequelae of kidney disease, which originates as tubular damage and then proceeds to reductions in eGFR after substantial nephron loss (in some estimates > 50% of nephron loss) has occurred. The authors must clarify throughout (kidney and repro tables) when studies reviewed are cross-sectional vs. longitudinal in design and when reverse causality limits conclusions regarding temporality.
Major:
Line 85 (repeated in line 439)–The scientific premise of this sentence is not appropriate, particularly considering the (lack of) temporal context of the preceding line/study referenced. “Thus, an elevation of β2MG excretion is not an early sign of nephrotoxicity of Cd” is a separate question from whether β2MG levels correlate with existing or cross-sectional CKD status as implied by “β2MG excretion rates of 100-299, 300-999 and ≥1000 μg/g creatinine were associated with 4.7-, 6.2 and 10.5-fold increases in risk of estimated glomerular filtration rate (eGFR) ≤ 60 mL/min/1.73m2, commensurate with CKD [18].” As reference 18 indicates, the inverse association between eGFR and urinary β2-MG was only observed among individuals with low eGFR (values below 60), and NOT among individuals with eGFR >90 (renally healthy individuals). Thus among individuals with extant CKD, defined by eGFR< 60, B2M is not a valid ‘early’ marker since disease is already present. However among healthy individuals it may still be relevant (as is shown in other prior studies with B2M as an outcome). The authors should expand their review scope to include B2M as an outcome of interest in order to make conclusions regarding the evidence of B2M as an indicator prior to the onset of eGFR decline. Since in the sequelae of kidney disease, we would consider “early sign” to mean, prior to the onset of eGFR decline and substantial nephron loss, this question was not addressed in reference 18, nor was this literature summarized, and therefore does not support the statements on line 85/439.
In addition to including a review of articles where B2M is the outcome. The authors could also address this in a re-analysis of data from [18] where eGFR >90, and examine the association between Cd and B2M in the absence of kidney function decline. Note that the conclusions would still be limited due to the cross-sectional design.
Lines 170-174 seem contradictory – the is a lack of excretory mechanisms and virtually all Cd is taken up by the kidney (line 170), yet urine provides the best estimate of kidney burden because the amount excreted varies (typo, line 173) with the amount accumulated. Can the authors clarify/ expand on this discrepancy?
Line 188-I disagree with the authors consideration that “Cd-induced reduction in GFR” is a suitable endpoint for risk calculation. As the authors state in the paragraph above, a loss in GFR would temporally occur after tubular injury, fibrosis, and impaired reabsorption which could sequentially reduce GFR, and lead to further retention impaired reabsorption of Cd with concurrent albuminuria indicating glomerular damage. This is precisely why B2M is considered an early marker of kidney dysfunction (prior to reduced GFR) among healthy individuals (or prior to eGFR decline in CKD).
Line 226 – it is critical to interpretation of the literature reviewed in Tables 1 & 2 and lines 212-225 whether the study designs are cross sectional or longitudinal and whether low eGFR exists alongside elevated Cd and therefore subject to reverse causation as an equally plausible conclusion.
Minor:
The official gene name abbreviation for beta 2 microglobulin is B2M: https://www.genecards.org/cgi-bin/carddisp.pl?gene=B2M
Line 36 – please elaborate why “cheap”?
Line 94/99 – and kidneys. (as suggested by line 122)
Line 116 – please include a reference
Line 127 – needs a reference and clarification on “toxicity becomes unacceptable”
Line 130 – NHANES is a cross-sectional study that does not address temporality; “seen before” and “reaches” should be toned down to “at UCd levels below 5.24”
Line 132- please describe the population examined in Ref 29 – sample size, age, did exposure occur among the general population?
Table 1 – Multiple targets can be replaced with “kidney, liver and pancreas”
Ref 31 & 33 – is a cross-sectional study (NHANES) therefore the word “increase” is not appropriate as it suggests temporality. (Table 1 & 2) (line 196); “deterioration” (line 215)
Typos:
Line 62 – typo “notable” should be ‘notably’
Line 167 – “is accumulate”
Author Response
We thank the reviewers for the constructive comments, advice, and the opportunity to improve our manuscript. Accordingly, it has now been extensively revised. Itemized responses to the reviewer’s comments are below. Changes made to the manuscript are in blue.
Response to Reviewer 1
Top comment
The manuscript is generally well-written and thorough with respect to reproductive organ review. However, there are major concerns regarding completeness of the kidney review and conclusions /scientific premises that are not substantiated (see specific comments regarding lines 85/439 and 188). The section 4.2 on reduced eGFR should also include B2M (and other kidney injury markers) as outcomes. The citation of a single study in reference 18 (Satarug et al 2019) does not sufficiently or justifiably support the authors’ conclusions that B2M is not an appropriate biomarker in the sequelae of kidney disease, which originates as tubular damage and then proceeds to reductions in eGFR after substantial nephron loss (in some estimates > 50% of nephron loss) has occurred. The authors must clarify throughout (kidney and repro tables) when studies reviewed are cross-sectional vs. longitudinal in design and when reverse causality limits conclusions regarding temporality.
Response:
- The kidney review section has been improved by the addition of four new subsections.
- Kidney injury markers NAG and KIM1 are include in section 4.3
- B2M is addressed in new section 4.4.
- The dose-response relationships between Cd exposure metrics and certain outcomes measures as revealed by the systematic and meta-analysis reviews have been added.
- The cross-sectional nature of NHANES studies is stated.
New subsections
- The Kidney as the Target of Cadmium Toxicity.
4.1. Accumulation of Cadmium in Human Organs
4.2. Reabsorption of Cadmium and Its Excretion
4.3. Kidney Tubular Cell Injury
4.4. Impaired Reabsorption of Filtered Proteins
.
Major point 1: Line 85 (repeated in line 439)–The scientific premise of this sentence is not appropriate, particularly considering the (lack of) temporal context of the preceding line/study referenced. “Thus, an elevation of β2MG excretion is not an early sign of nephrotoxicity of Cd” is a separate question from whether β2MG levels correlate with existing or cross-sectional CKD status as implied by “β2MG excretion rates of 100-299, 300-999 and ≥1000 μg/g creatinine were associated with 4.7-, 6.2 and 10.5-fold increases in risk of estimated glomerular filtration rate (eGFR) ≤ 60 mL/min/1.73m2, commensurate with CKD [18].” As reference 18 indicates, the inverse association between eGFR and urinary β2-MG was only observed among individuals with low eGFR (values below 60), and NOT among individuals with eGFR >90 (renally healthy individuals). Thus among individuals with extant CKD, defined by eGFR< 60, B2M is not a valid ‘early’ marker since disease is already present. However among healthy individuals it may still be relevant (as is shown in other prior studies with B2M as an outcome). The authors should expand their review scope to include B2M as an outcome of interest in order to make conclusions regarding the evidence of B2M as an indicator prior to the onset of eGFR decline. Since in the sequelae of kidney disease, we would consider “early sign” to mean, prior to the onset of eGFR decline and substantial nephron loss, this question was not addressed in reference 18, nor was this literature summarized, and therefore does not support the statements on line 85/439.
In addition to including a review of articles where B2M is the outcome. The authors could also address this in a re-analysis of data from [18] where eGFR >90, and examine the association between Cd and B2M in the absence of kidney function decline. Note that the conclusions would still be limited due to the cross-sectional design.
Response:
- Reports of use of β2M as the outcome are provided in Section 4.4.
- An increase in β2M excretion reflects an increase in its production due to inflammatory and neoplastic conditions or a substantial reduction in GFR. These two attributes compromise its utility as a marker of tubulopathy.
- Current evidence suggests that KIM1 is the most sensitive marker of the toxicity of Cd to kidneys (Section 4.3).
- We acknowledge the suggestion to re-analysis of our data for an association between β2M and Cd among those who had a GFR in the normal range. This will be included in our future work.
Major point 2.
Lines 170-174 seem contradictory – the is a lack of excretory mechanisms and virtually all Cd is taken up by the kidney (line 170), yet urine provides the best estimate of kidney burden because the amount excreted varies (typo, line 173) with the amount accumulated. Can the authors clarify/ expand on this discrepancy?
Response:
- A slow excretion rate for Cd is clarified in subsection 4.1. This was from reports of Swedish autopsy studies [ref. 49,50].
- The amount of excreted Cd in relation to kidney burden is discussed in 4.2.
4.1. Accumulation of Cadmium in Human Organs
4.2. Reabsorption of Cadmium and Its Excretion
Major point 3.
Line 188-I disagree with the authors consideration that “Cd-induced reduction in GFR” is a suitable endpoint for risk calculation. As the authors state in the paragraph above, a loss in GFR would temporally occur after tubular injury, fibrosis, and impaired reabsorption which could sequentially reduce GFR, and lead to further retention impaired reabsorption of Cd with concurrent albuminuria indicating glomerular damage. This is precisely why B2M is considered an early marker of kidney dysfunction (prior to reduced GFR) among healthy individuals (or prior to eGFR decline in CKD).
Response:
- Many reports of GFR effect of Cd are added to Section 4.5. Cadmium-Induced GFR Reduction.
- The pathogenesis of GFR decrease due to Cd nephropathy is explained. Key statements are quoted below.
“Reductions in GFR due to Cd nephropathy has often been attributed to glomerular injury. Although this inference may be at least partially correct, it is not necessary. Sufficient tubular injury disables glomerular filtration and ultimately leads to nephron atrophy, glomerulosclerosis, and interstitial inflammation and fibrosis [Ref. 83].
Ref. 83. Schnaper, H.W. The tubulointerstitial pathophysiology of progressive kidney disease. Adv. Chron. Kidney Dis. 2017, 24, 107-116.
- A fall in GFR is an early response to Cd toxicity. Excreted Cd is an indicator of tubular injury itself because the excreted Cd originates from injured or dying tubular cells [Ref. 61].
- Excretion of β2M reflects a decrease in tubular reabsorption because most excreted β2M are from plasma, not from tubular cells.
- A risk assessment model that assumes β2M excretion above ≥300 µg/ g creatinine as a critical endpoint yielded a toxicity threshold at urinary Cd excretion rate of 5.24 µg/ g creatinine.
- Many studies consistently observed associations between of urinary Cd excretion rates below 5 than 5.24 µg/ g creatinine and GFR < 60mL/min.1,73 m2 and albuminuria.
Major point 4.
Line 226 – it is critical to interpretation of the literature reviewed in Tables 1 & 2 and lines 212-225 whether the study designs are cross sectional or longitudinal and whether low eGFR exists alongside elevated Cd and therefore subject to reverse causation as an equally plausible conclusion.
Response:
- We have specified a study design and where applicable with wording changes.
- We have added a dose-response analysis that is available for diabetes outcome [Ref. 42].
- BMDL values available for type 2 diabetes outcome have been provided [Ref. 38]
Minor issues
The official gene name abbreviation for beta 2 microglobulin is B2M: https://www.genecards.org/cgi-bin/carddisp.pl?gene= β2M
Response:
- β2MG has been changed to β2M throughout our paper.
Line 36 – please elaborate why “cheap”?
Response:
- The referred word “cheap” has been deleted.
Line 94/99 – and kidneys. (as suggested by line 122)
Response:
- The kidney has been added.
- Cd accumulation in kidneys is detailed in 4.1. Accumulation of Cadmium in Human Organs.
- A new Figure (Figure 2) has been added to Section 4.2.
Line 116 – please include a reference
Response: Ref. 23.
Line 127 – needs a reference and clarification on “toxicity becomes unacceptable”
Response: Ref. 14.
Line 130 – NHANES is a cross-sectional study that does not address temporality; “seen before” and “reaches” should be toned down to “at UCd levels below 5.24”
Response:
- Wording has been changed.
Line 132- please describe the population examined in Ref 29 – sample size, age, did exposure occur among the general population?
Response.
- Information sought has been provided (lines 133-136).
Table 1 – Multiple targets can be replaced with “kidney, liver and pancreas”
Ref 31 & 33 – is a cross-sectional study (NHANES) therefore the word “increase” is not appropriate as it suggests temporality. (Table 1 & 2) (line 196); “deterioration” (line 215)
Response:
- Rewording where applicable.
Typos:
Line 62 – typo “notable” should be ‘notably’
Line 167 – “is accumulate”
Response:
- Typos have been corrected.

Reviewer 2 Report
· The abstract lacks innovation and appears more like a synopsis. The writers must make it apparent how their work helps close the gap between published work and emerging trends. The research’s objective, key findings, and main conclusions should all be briefly stated in the abstract. It must be able to stand alone since an abstract is frequently offered apart from the article.
· It is advised to outline the article’s structure towards the conclusion of the introduction. Comparing the current study’s findings with some earlier related research is advised.
· The authors are advised to check the grammatical errors throughout the manuscript.
· The subsections of this review do not properly connect. To support the text, more information and new case studies are needed.
· More pictorial representations are required to express the idea conveyed in the manuscript.
· The conclusion section does not correctly address the important points and future perspectives. The author should add a few important points.
· It has been observed that proper referencing is missing throughout the manuscript. Additionally, it has been observed that large chunks of data have been extracted from single papers without proper citations.
· Figure 1: ‘Multiple toxicity targets for cadmium’ is difficult to understand. The authors are advised to modify the figure showing organ toxicity in response to Cd exposure.
· Try to reduce the figure legend’s length and avoid too much-written data in the figures.
· The authors are advised to demonstrate the pathophysiology of cadmium toxicity concerning the specific organs in detail.
· Why the authors focused only on a low cadmium dose is still unclear. Proper explanations or instances should be described how the transition from low to high doses of cadmium results in several disorders and toxicity. Additionally, a proper explanation should be given about the acute and chronic effects of cadmium exposure.

Author Response
Response to Reviewer 2
We thank the Reviewer for comments and advice for improving our manuscript. We have revised our paper extensively. We provide below point-by-point response to issues/concerns raised by the Reviewer. Changes made to the text in a manuscript are in blue.
Point 1. The abstract lacks innovation and appears more like a synopsis. The writers must make it apparent how their work helps close the gap between published work and emerging trends. The research’s objective, key findings, and main conclusions should all be briefly stated in the abstract. It must be able to stand alone since an abstract is frequently offered apart from the article.
Response:
- The abstract has been rewritten to reflect new knowledge synthesis unique to our paper (lines 12-24).
Point 2. It is advised to outline the article’s structure towards the conclusion of the introduction. Comparing the current study’s findings with some earlier related research is advised.
Response:
- The last paragraph of an Introduction has been rewritten to link information in previous paragraphs to the aims of our review (lines 56-62).
Point 3. The authors are advised to check the grammatical errors throughout the manuscript.
Response:
- The English grammatical errors have been checked.
Point 4. The subsections of this review do not properly connect. To support the text, more information and new case studies are needed.
Response:
- Section 3 has a new title “Organs Susceptible to Cadmium Toxicity”.
- Two subsections are added to this Section 3
Fate of Cadmium in the Body.
Target Organ Toxicity Evident from U.S. NHANES.
- Four new subsections are added to Section 4.
Accumulation of Cadmium in Human Organs
Reabsorption of Cadmium and Its Excretion
Kidney Tubular Cell Injury
Impaired Reabsorption of Filtered Proteins
- Many more studies in humans from over 25 papers are added.
Point 5. More pictorial representations are required to express the idea conveyed in the manuscript.
Response:
- A Figure 2 has been inserted to explain a high Cd accumulation level in kidney tubular cells.
Point 6. The conclusion section does not correctly address the important points and future perspectives. The author should add a few important points.
Response:
- The conclusion has been rewritten to recapitulate the important points and population health implications (lines 532-547).
Point 7. It has been observed that proper referencing is missing throughout the manuscript. Additionally, it has been observed that large chunks of data have been extracted from single papers without proper citations.
Response:
- Please see responses to Point 4 above.
Point 8. Figure 1: ‘Multiple toxicity targets for cadmium’ is difficult to understand. The authors are advised to modify the figure showing organ toxicity in response to Cd exposure.
Response
- Figure 1 and its legend has been simplified as advised.
Point 9. Try to reduce the figure legend’s length and avoid too much-written data in the figures.
Response:
- The legend to Figure 1 has been shortened.
Point 10. The authors are advised to demonstrate the pathophysiology of cadmium toxicity concerning the specific organs in detail.
Response:
- Current knowledges on the pathophysiology of Cd toxicity in liver and gonads are still fragmentary. The pathogenesis of the hepatoxicity and gonadotoxicity of Cd cannot be constructed.
- The pathogenesis of GFR reduction due to Cd nephropathy is discussed in Section (lines A new Figure 2 has been added.
Point 11. Why the authors focused only on a low cadmium dose is still unclear. Proper explanations or instances should be described how the transition from low to high doses of cadmium results in several disorders and toxicity. Additionally, a proper explanation should be given about the acute and chronic effects of cadmium exposure.
Response:
- A new section 3 should rectify the concerns raised. It is self-evident that studies in humans reported here are chronic environmental exposure situations.
- Cd exposure duration and dose-regimens are provided in sufficient details. In describing the experimental data pertaining to the gonadotoxicity of Cd.

Round 2
Reviewer 1 Report
Thanks to the authors for these substantial additions as well as inclusion of more nuanced and tempered language surrounding their recommendations.
I request a few additional important and nuanced considerations below.
In the response Ref 14 was provided for the minor concern “Line 127 – needs a reference and clarification on “toxicity becomes unacceptable””
Thank you for providing the reference. Please note that the embedded link does not route to the pdf properly. Page 159 of the WHO report notes that the breakpoint is derived for a population over 50 years of age – which is an important point to add regarding the threshold derivation. Please note that the >300 ug/g creatinine is not a “rate” (as written) but a level measured at a single time point. Incorrect use of the word ‘rate’ should be corrected throughout. Similarly, a reference to a B2M ‘rate’ occurs throughout.
It appears that the threshold was not established based on ‘toxicity’ being acceptable or unacceptable, but rather that urinary levels below 5.24 ug Cd/g creatinine were not associated with a change in B2M excretion. An argument could also be made that B2M is not sensitive enough to observe changes below this threshold of Cd. The age of participants used to derive this breakpoint is also important as B2M can vary by age (increases), and Cd levels can vary by age (increases). The authors should note both as alternate rationales for why a revised threshold level derivation may be warranted based on new data.
https://www.ncbi.nlm.nih.gov/pmc/articles/PMC4529371/ (in plasma of mice)
I would be curious too, if the authors’ analysis of their own B2M data show a correlation between urinary B2M and age (of renally healthy individuals).
Beginning on page 5 and onward, the authors should be cautious to use “eGFR” as appropriate when it has been estimated (as in NHANES an many observational studies) versus “GFR” which can indicate that is was measured by iohexol – typically abbreviated as mGFR.
Line 296 – “over the last 17 years” is misleading since each cycle of NHANES is cross-sectional and individuals are not followed over time. The issue of reverse causality of studies where reduced eGFR is already present and Cd levels are higher (both confounded by age) is an important point that must be noted in the discussion of whether eGFR is a suitable endpoint of toxicity. (as in lines 316-317)
Line 305 – this study is also cross-sectional. Please add.
In the response, authors state “A fall in GFR is an early response to Cd toxicity.” Excreted Cd is an indicator of tubular injury itself because the excreted Cd originates from injured or dying tubular cells [Ref. 61]. Reference 61 does not adequately support the assertion that eGFR decline is an “early” response to Cd toxicity. Section 4.5 does not indicate strong evidence to support this, as very few studies have been conducted showing the Cd levels measured before eGFR decline have predictive ‘early’ responsiveness. In nearly all cases decreased eGFR is measured at the same time as Cd levels, meaning that temporal assessments of “early” cannot be established. This is the issue of reverse causality which has affected most studies of CKD and eGFR relationships and must be noted as a major limitation. As the authors indicate in their updated manuscript and response, KIM-1 is suggested as a reliable early indicator of Cd nephrotoxicity. Lines 538-540 should be revised in light of the issues raised.
Author Response
Replies to additional comments
We thank the Reviewer for her/his insightful comments and a thorough review of our manuscript.
Comment 1
In the response Ref 14 was provided for the minor concern “Line 127 – needs a reference and clarification on “toxicity becomes unacceptable””
Reply:
- With reference to “toxicity becomes unacceptable” phrase, it has been deleted and sentences have been changed according to advice given (lines 122-130).
Thank you for providing the reference. Please note that the embedded link does not route to the pdf properly. Page 159 of the WHO report notes that the breakpoint is derived for a population over 50 years of age – which is an important point to add regarding the threshold derivation. Please note that the >300 ug/g creatinine is not a “rate” (as written) but a level measured at a single time point. Incorrect use of the word ‘rate’ should be corrected throughout. Similarly, a reference to a B2M ‘rate’ occurs throughout.
Reply:
- Correction on B2MG has been undertaken where required.
- Age consideration in a risk assessment model of JECFA and EFSA has been inserted (line 78).
It appears that the threshold was not established based on ‘toxicity’ being acceptable or unacceptable, but rather that urinary levels below 5.24 ug Cd/g creatinine were not associated with a change in B2M excretion. An argument could also be made that B2M is not sensitive enough to observe changes below this threshold of Cd. The age of participants used to derive this breakpoint is also important as B2M can vary by age (increases), and Cd levels can vary by age (increases). The authors should note both as alternate rationales for why a revised threshold level derivation may be warranted based on new data.
Reply:
- Logical arguments suggested have been inserted (line 321-326).
https://www.ncbi.nlm.nih.gov/pmc/articles/PMC4529371/ (in plasma of mice)
I would be curious too, if the authors’ analysis of their own B2M data show a correlation between urinary B2M and age (of renally healthy individuals).
Reply:
- In our paper submitted to IJERPH, “Dose-Response Analysis of the Tubular and Glomerular Effects of Chronic Exposure to Environmental Cadmium” we reported that B2M excretion was weakly associated with age (β = 0.078, p = 0.038) in a multiple regression analysis including all subjects.
- We will examine in our future work the associations of B2MG and age in subjects stratified by eGFR levels <60, 61-89, ≥ 90 mL/min/1.73m2.
Beginning on page 5 and onward, the authors should be cautious to use “eGFR” as appropriate when it has been estimated (as in NHANES an many observational studies) versus “GFR” which can indicate that is was measured by iohexol – typically abbreviated as mGFR.
Reply:
- Explanations regarding GFR and eGFR have been inserted along with a reference #84 (lines 264-268). eGFR has been used as specified.
Line 296 – “over the last 17 years” is misleading since each cycle of NHANES is cross-sectional and individuals are not followed over time. The issue of reverse causality of studies where reduced eGFR is already present and Cd levels are higher (both confounded by age) is an important point that must be noted in the discussion of whether eGFR is a suitable endpoint of toxicity. (as in lines 316-317)
Reply:
- A statement appearing in line 296 has now been reworded.
Line 305 – this study is also cross-sectional. Please add.
Reply:
In the response, authors state “A fall in GFR is an early response to Cd toxicity.” Excreted Cd is an indicator of tubular injury itself because the excreted Cd originates from injured or dying tubular cells [Ref. 61]. Reference 61 does not adequately support the assertion that eGFR decline is an “early” response to Cd toxicity. Section 4.5 does not indicate strong evidence to support this, as very few studies have been conducted showing the Cd levels measured before eGFR decline have predictive ‘early’ responsiveness. In nearly all cases decreased eGFR is measured at the same time as Cd levels, meaning that temporal assessments of “early” cannot be established. This is the issue of reverse causality which has affected most studies of CKD and eGFR relationships and must be noted as a major limitation. As the authors indicate in their updated manuscript and response, KIM-1 is suggested as a reliable early indicator of Cd nephrotoxicity.
Lines 538-540 should be revised in light of the issues raised.
Reply:
- With reference to a GFR statement, it should read, “It appeared that GFR may fall before β2M excretion increases to level above 300 µg/g creatinine.”
- A below statement on KIM-1 has been inserted (lines 549-551).
“KIM1 has emerged as an early warning sign of Cd nephrotoxicity and its use in conjunction with β2M, albumin and eGFR in a longitudinal study design warrants further research”.

Reviewer 2 Report
Accept
Author Response
We thank the Reviewer for an evaluation of our paper.